# ZnO:V Nanoparticles with Enhanced Antimicrobial Activities

**Leila Alaya [1], Ahmad Mohammad Saeedi [2], Ahmad Abdulhadi Alsaigh [3], Meshal H. K. Almalki [3,4], Norah Hamad Alonizan [5] and Mokhtar Hjiri [6,***

1 Department of Life Sciences, Faculty of Sciences of Gabes, Gabes 6072, Tunisia; leila_alaya@yahoo.fr
2 Department of Physics, Faculty of Applied Science, Umm AL-Qura University, Makkah 24382, Saudi Arabia; amasaeedi@uqu.edu.sa
3 Department of Biology, Faculty of Applied Science, Umm Al-Qura University, Makkah 24382, Saudi Arabia; aassaigh@uqu.edu.sa (A.A.A.); mhmaleki@uqu.edu.sa (M.H.K.A.)
4 Research Laboratories Unit, Faculty of Applied Science, Umm Al-Qura University, Makkah 24382, Saudi Arabia
5 Department of Physics, College of Science, Imam Abdulrahman bin Faisal University, P.O. Box 1982, Dammam 31441, Saudi Arabia; nalonizan@iau.edu.sa
6 Department of Physics, College of Sciences, Imam Mohammad ibn Saud Islamic University (IMSIU), Riyadh 11623, Saudi Arabia
* Correspondence: mbhjiri@imamu.edu.sa or m.hjiri@yahoo.fr; Tel.: +96-65-0616-3909

**Abstract:** In this study, we used sol-gel to synthesize undoped and V-ZnO nanoparticles with different vanadium concentrations (1, 3, and 5 at.%) under supercritical dry conditions of ethanol. XRD spectra showed that the obtained powders are well crystallized in the hexagonal wurtzite structure of ZnO nanoparticles. The average crystallite size, estimated by the Debye-Scherer formula, was found to be equal to 31 nm for the pure sample, and it was decreased to 27 nm for the 3at.% vanadium-doped one. SEM and TEM photographs indicated the spherical and elongated shapes of the nanoparticles. The stretching bands located at 419 cm$^{-1}$ confirmed ZnO material formation. The efficacy of the produced ZnO NPs against Gram$^+$, Gram$^-$ bacteria, and fungi was tested. Vanadium-doped ZnO, with low concentrations (10 µg/mL), exhibited a large influence on bacterial and fungi growth inhibition. For example, the inhibition zones IZ of *S. aureus* and *E. coli* bacteria reached 16 and 15 mm, respectively, for ZnO:V$_{1\%}$, while the IZ of these two bacteria were 14 and 12 mm for the undoped ZnO. The use of V-dopant enhanced the production of the reactive oxygen species ROS by the photogeneration of electron-hole pairs due to light absorption by ZnO in the visible region.

**Keywords:** ZnO NPs; sol-gel; vanadium; dopant; bacteria; fungi

## 1. Introduction

In the last decades, several scientific researchers have given interest to microbial and bacterial contaminations because this pest is becoming a serious and hazardous problem in the food industry, waterborne, and healthcare [1]. In fact, these infections were among the first causes of mortality all over the world. Bacteria are microorganisms with a cell structure simpler than that of many other organisms. Their control center, containing the genetic information, is contained in a single loop of DNA. Some bacteria have an extra circle of genetic material called a plasmid rather than a nucleus. The plasmid often contains genes that give the bacterium some advantage over other bacteria. For example, it may contain a gene that makes the bacterium resistant to a certain antibiotic. *Staphylococcus aureus* can cause illness through preformed toxin production as well as by infecting both local tissues and the systemic circulation. *Shigella flexneri* causes more than one million deaths every year because of food contamination by this type of bacteria.

The use of different resistance drugs for a lot of microbes increased the risk of this issue. Every year, an important percentage of hospitalized patients are infected by drug-resistant bacteria [2]. In addition, overuse of antibiotics causes bacteria to become resistant

to them. In addition to this issue, when bacteria are in a certain environment, they tend to communicate among themselves and create a biofilm (protective layer). Polysaccharides forming the biofilm do not allow antibiotics to penetrate into the biofilm. Bacteria in a biofilm are extremely hard to kill. These circumstances prompted scientists to search for alternative solutions such as the use of organic (alcohols, chlorine, etc.) and inorganic antimicrobial agents that killed or slowed the microbes growth without being toxic or dangerous to the surrounding tissue. In comparison with organic agents, inorganic ones are more stable and can withstand higher temperatures and pressures.

Among the inorganic antimicrobial agents, metal oxide nanomaterials have been largely used in antibacterial activities to kill or inhibit bacteria growth. Eduardo's team has utilized $Ag/SnO_2$ composites against *E. coli* bacteria [3]. Rajagopalachar et al. synthesized MgO nanoparticles as an antibacaterial agent to kill *Bacillus cereus* and *Pseudomonas aeruginosa* bacteria [4]. Parc and his colleagues have used $CuO/TiO_2$ nanowires to inhibit the growth of *Escherichia coli* (*E. coli*) bacteria [5]. Alghamdi et al. have tested $Co_2SnO_4$ particles against both $Gram^+$ and $Gram^-$ bacteria [6].

Additionally, ZnO nanostructures have been synthesized using numerous techniques and processes. El Mir et al. have produced a new protocol for the preparation of ZnO nanoparticles using the sol-gel route [7]. Using the electrodeposition approach, ZnO nanoflowers and nanosheets have been synthesized by Kou and his team [8]. On the other hand, Wahab and his colleagues have produced ZnO with nanowire shapes using a low-cost and safe technique named the green chemistry process [9]. Additionally, the same approach was used to prepare ZnO with a spherical shape using citrus sinensis extract [10]. ZnO, with its different shapes, has been applied in several research domains. Hjiri et al. have tested ZnO nanoparticles against several hazardous gases, such as carbon monoxide CO and nitrogen dioxide NO2, to find promising gas sensors based on spherical-shaped ZnO materials [11,12]. El Mir and his team have produced a solar cell based on ZnO thin film as a transparent window [13]. Farzinfar and his colleagues have applied this material to the dye's degradation and photocatalysis [14]. Another application of ZnO was the light-emitting diode performed by Zhang and his team [15]. Daniel et al. have synthesized In doped ZnO with a spherical shape for antibacterial activities [16], which is the goal of our present work.

Numerous scientific researchers have considered ZnO as a promising inorganic nano-material used to produce an antimicrobial activity agent [17–20] due to its abundance in nature, low cost, non-toxicity, and chemical stability. Zinc oxide exhibited n-type conductivity, a wide band gap of 3.37 eV, and an excitonic binding energy equal to 60 meV at ambient temperature [21,22]. Furthermore, its great physical properties lead this material to find antimicrobial activity against several pathogenic microbial strains, including $Gram^+$ bacteria such as *Bacillus cereus* and *Staphylococcus aureus*, $Gram^-$ bacteria such as *Escherichia coli* and *Pseudomonas aeruginosa*, and fungi such as *Candida albicans* and *Rhodotorula glutinis* [23–25].

Despite its interesting properties, ZnO still suffers from a lack of efficiency against bacteria and fungi, especially in the visible region of light because of the wide band gap. As a solution, diverse approaches are followed to enhance the antimicrobial activities of ZnO nanomaterials. One of these ways is that doping ZnO with suitable metals improves antibacterial/anti-fungal activities and growth inhibition. The use of cadmium [26], ytterbium [27], nickel [28], cobalt [29], magnesium [30], boron [31], and tellurium [32] as dopants for ZnO nanoparticles has been investigated in the literature.

Doping ZnO with vanadium enhances the antibacterial activity of this material by generating reactive oxygen species (ROS). Pure ZnO suffers from the weak usage rate of visible radiation as a bactericidal agent because of its large band gap, resulting in low generation of ROS and bad antibacterial activity. Herein comes the role of the vanadium element to increase the light absorption of ZnO in the visible range by reducing the band gap and then enhance the photogeneration of electron-hole pairs, leading to amelioration of the production of ROS and, therefore, an increase in the antibacterial activity of ZnO nanopowders. Furthermore, the addition of V dopant in ZnO influences the crystallite

size by reducing it, causing damage to bacteria and fungi walls, and then enhancing the antibacterial activities. The vanadium element also helps in releasing $Zn^{2+}$ from ZnO material and then promotes ROS production.

In this context, the utilization of vanadium ions for doping ZnO nanomaterials was investigated. The influence of this dopant on structure and morphology was studied. Furthermore, the effect of these ions on the amelioration of the antimicrobial activities of this material against different pathogenic strains has been largely demonstrated. The different mechanisms involved in bacterial activity are well investigated and explained.

## 2. Materials and Methods

### 2.1. V-ZnO Powders Synthesis

The sol-gel technique was performed to obtain pure ZnO and V-doped ZnO nanopowder samples with different loads of vanadium (1, 3, and 5 at.%). One dissolved 16 g of zinc precursor [$Zn(OOCCH_3, 2 H_2O)$; 98%] in 112 milliliters of methyl alcohol [$CH_3OH$], then added the quantity of ammonium metavanadate [$NH_4VO_3$], which corresponds to each vanadium doping concentration (1, 3, and 5 at.%), and continued stirring until the solubility of all the precursors and a clear solution were obtained. The obtained solution was poured into a stainless-steel autoclave and dried in supercritical conditions (ethyl alcohol as co-solvent; Tc = 243 °C; Pc = 63.6 bars) so that powdered aerogels could be obtained [7]. Finally, the samples are calcinated for 2 h at 500 °C in air using a furnace. Sample codes are written as V0ZO, V1ZO, V3ZO, and V5ZO, according to the nominal vanadium load of each sample. Some quantities of prepared powders were used to do different characterizations (XRD, scanning electron microscopy (SEM), energy dispersive X-ray spectroscopy (EDX), Fourier transform infrared (FTIR) and the rest of the powders were tested as antimicrobial agents against Gram$^+$ and Gram$^-$ bacteria strains and two kinds of fungi.

### 2.2. Characterization

The X-ray diffraction apparatus of type Bruker AXS D8 Advance uses 1.5405 Å as $CuK_{\alpha 1}$ wavelength. We have estimated the average particle size using Scherrer's formula:

$$G = \frac{0.9\lambda}{B \cos \theta_B} \tag{1}$$

From Formula (1), the X-ray wavelength is $\lambda$, $B$ is the full width at half maximum (FWHM) of the XRD peak, and the maximum of the Bragg diffraction peak (in radians) is noted $\theta_B$.

The shape and size of the synthesized nanopowders were shown by transmission electron microscopy (TEM) carried out with an electron microscope of type JEOL JEM 2010 ($LaB_6$ electron gun, JEOL, Tokyo, Japan) that operates at 200 kV and is equipped with a Gatan 794 Multi-Scan CCD camera to do digital imaging. The surface of samples is observed using SEM observations performed using an instrument of the type Zeiss Cross Beam 540. The absorption coefficients from Fourier transform infrared (FTIR) were carried out with a spectrophotometer of type Brucker and ranged from 400 to 4000 cm$^{-1}$.

### 2.3. Antibacterial Assay

#### 2.3.1. Microbial Strains

Six pathogenic microbial strains, including two Gram$^+$ bacteria (*Bacillus cereus* ATCC 11,778 and *Staphylococcus aureus* ATCC 25923), two Gram$^-$ bacteria (*Escherichia coli* ATCC 25,922 and *Pseudomonas aeruginosa* ATCC 25668), and two strains of opportunistic fungal (*Candida albicans* and *Rhodotorula glutinis*), were gotten from the culture collection in the Biology Department, College of Applied Sciences, The University of Umm Al-Qura, Makkah, Saudi Arabia. The initial cultures of microbial strains were kept at 4 °C on nutrient and sabouraud agar slants, respectively.

### 2.3.2. Assessment of Antimicrobial Activities

The antimicrobial activities of four different ZnO powders were assessed using the well-diffusion technique. Sterilized petri dishes were prepared with sterilized nutrient agar media for the application of bacteria strains, whereas sterilized sabouraud agar media was applied for fungi tests. All media were left to solidify in petri dishes at room temperature ($25 \pm 1$ °C) before being inoculated with fresh bacterial or fungal strain cultures. By using a sterile cork borer (8 mm diameter), wells were performed in the agar plates. Each well was filled with 100 µL of different ZnO NPs components suspended in dimethyl sulfoxide (DMSO) at a concentration of 10 µg/mL for each component. All plates were kept at ambient temperature for four hours to allow diffusion before incubation at 37 °C for 48 h. The potential antimicrobial activities of different components of ZnO NPs were stated when the inhibition zone diameter was measured for every well, with DMSO stated as a positive control and distilled water as a negative control for each microbial strain.

## 3. Results and Discussion

### 3.1. Samples Microstructure

X-ray diffractometers are designed for obtaining the highest quality diffraction data, combined with ease of use and flexibility to quickly switch to different applications. The XRD spectra of undoped ZnO and V-ZnO NPs are observed in Figure 1. This technique is non-destructive and is utilized to calculate the crystallite size and determine the structure of the materials. All diffraction peaks, located at $2\theta = 31.68, 34.40, 36.23, 47.55, 56.56, 62.90, 67.92$, and $69.09°$, corresponding to Miller indices (100), (002), (101), (102), (110), (013), (112), and (021), respectively, emphasized that the prepared samples are well crystallized in the structure called hexagonal Wurtzite, as indicated in the JPCDS card N° 36-1451 [33]. The non-existence of secondary phases was evidence for the high purity of the synthesized nanomaterials and the efficacy of the sol-gel technique. Most ZnO nanoparticles are preferentially oriented in the (101) direction. This result was also noticed with In-doped ZnO [16], Ca-doped ZnO [34], and Al-doped ZnO [12]. The average crystallite size of all samples was determined by Sherrer's formula [35]. As illustrated in Table 1, the particle size decreased with 1 at.% vanadium dopant addition from 31 to 26 nm and increased for a 3 at.% doped sample compared to a 1 at.% doped one. Adding dopant concentration until 5 at.% raised the particle size, but it was still less than the undoped one. The reduction in crystallite size was probably due to an increase in nucleation centers during the nanomaterial synthesis [36]. In contrast, the enhancement of the size may be due to the creation of agglomerates with the increase in dopant load, which leads to larger particles [37].

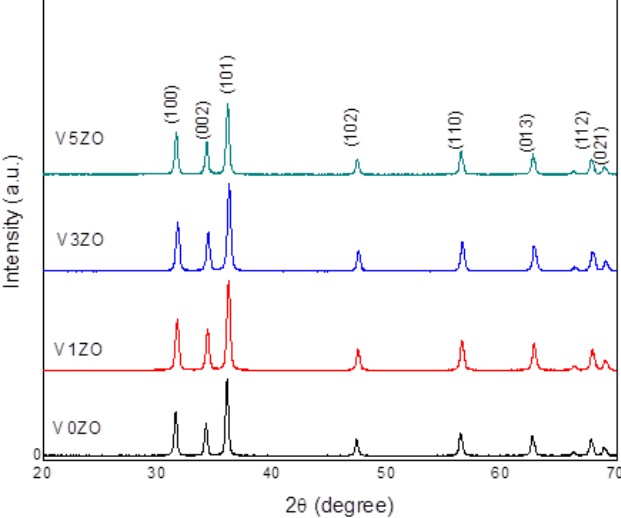

**Figure 1.** XRD patterns of undoped and vanadium modified ZnO nanopowders.

**Table 1.** Structural data reported from XRD analyses (101 plane).

| Samples | I-Max (Counts) | 2θ (deg) | B (FWHM) (deg) | B (FWHM) (Radian) | G (nm) | d (A) |
|---------|----------------|----------|----------------|-------------------|--------|-------|
| V0ZO | 2109 | 36.16 | 0.2725 | 0.0048 | 31 | 2.4878 |
| V1ZO | 2511 | 36.28 | 0.3240 | 0.0057 | 26 | 2.4792 |
| V3ZO | 2430 | 36.20 | 0.3182 | 0.0056 | 27 | 2.4760 |
| V5ZO | 1981 | 36.04 | 0.2917 | 0.0051 | 29 | 2.4880 |

The Fourier transform infrared spectroscopy (FTIR) recorded the vibrational bonds of undoped and V-ZnO nanopowders. The obtained FTIR patterns are reported in Figure 2. A stretching band at 419 cm$^{-1}$ was attributed to ZnO material. Another band observed at 940 cm$^{-1}$ was due to C-H vibrations. The weak band at 620 cm$^{-1}$ observed only for doped samples is assigned to vanadium oxide [38].

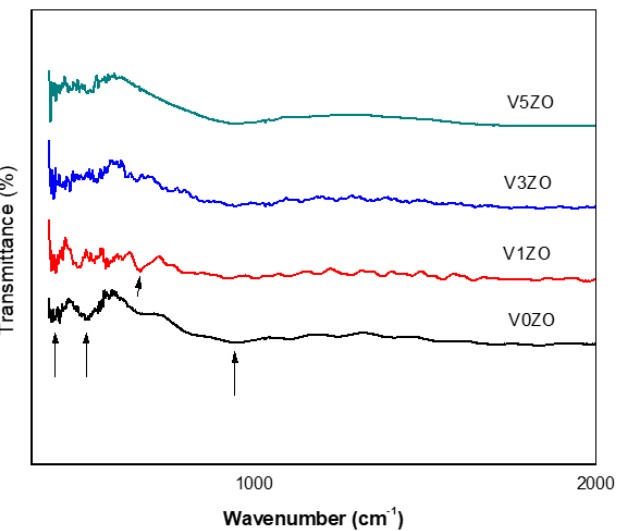

**Figure 2.** FTIR patterns of undoped and V-ZnO nanoparticles.

*3.2. Samples Morphology*

To confirm nanoparticle size obtained from XRD analysis by the Sherer formula, scanning electron microscopy (SEM) and scanning transmission microscopy (TEM) observations are made on undoped and V-ZnO nanoparticles.

Figure 3 shows SEM photographs of undoped and V-ZnO samples calcinated at 500 °C for two hours. Pure ZnO exhibited traditional grains with a spherical and regular shape. The grain size was estimated to be less than 1 μm. Incorporating vanadium dopant in the ZnO network caused agglomerates to appear, and the grains became larger with homogeneous size, as seen in the images. EDX spectra indicated the presence of the main elements Zn, O, and V without any other impurities, indicating the purity of the obtained materials. The element percentages are reported in Table 2. Zinc exhibited a percentage of 45.59 at.%, oxygen had a percentage of 52.2 at.%, and the percentage of vanadium was 2.21 at.%.

TEM images of pure and ZnO:V3% nanoparticles heat-treated at 500 °C for 2 h are reported in Figure 4. The undoped sample was composed of particles having a hexagonal shape and some other particles with an irregular shape. For the 3 at.% vanadium-doped ZnO, the particle shape was modified. In fact, they exhibited a prismatic shape and irregular size. The crystallite sizes were in good agreement with those obtained by XRD analyses.

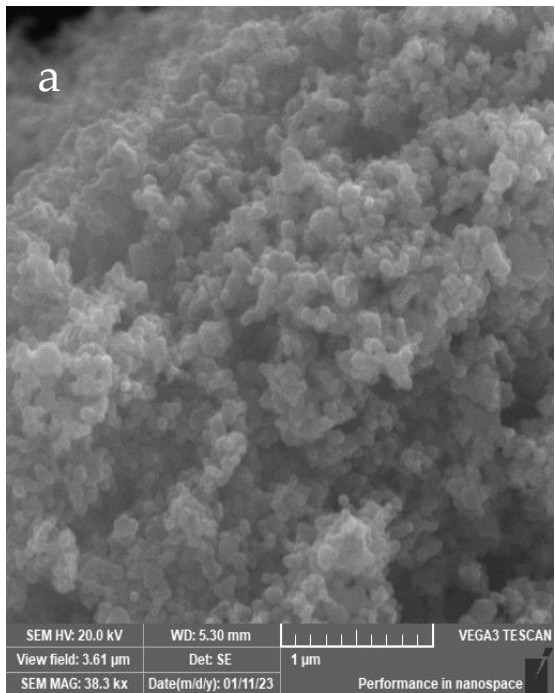
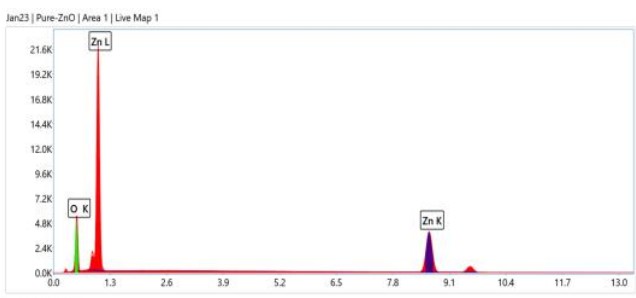
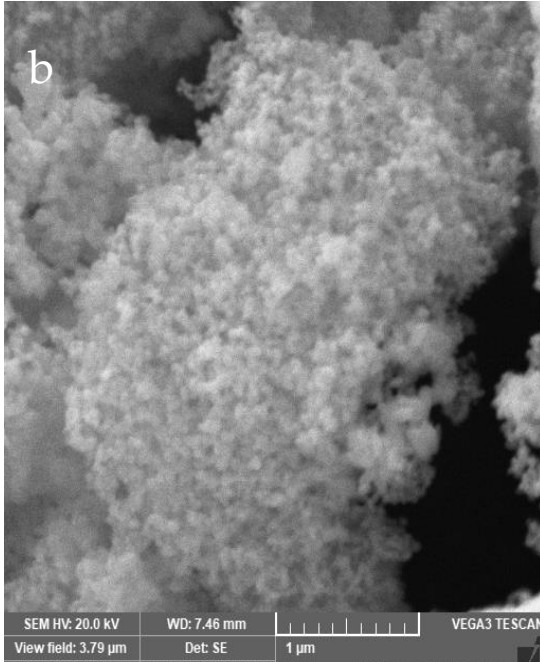
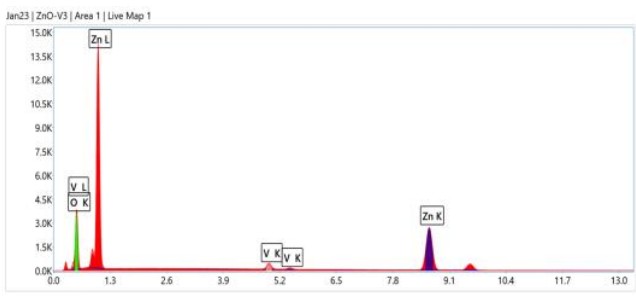

**Figure 3.** SEM photographs and EDX spectra: (**a**) Pure ZnO and (**b**) ZnO:V3%.

**Table 2.** Different elements percentages of V3ZO samples from EDX technique.

| Element | Weight % | Atomic % |
|---|---|---|
| O K | 21.26 | 52.2 |
| V K | 2.87 | 2.21 |
| Zn K | 75.87 | 45.59 |

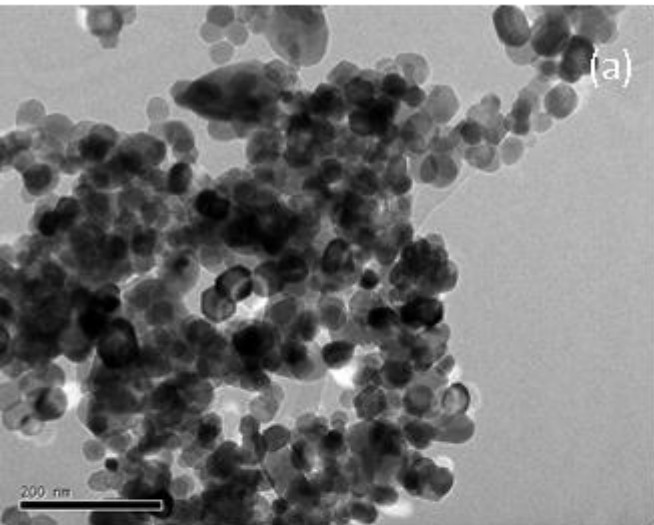

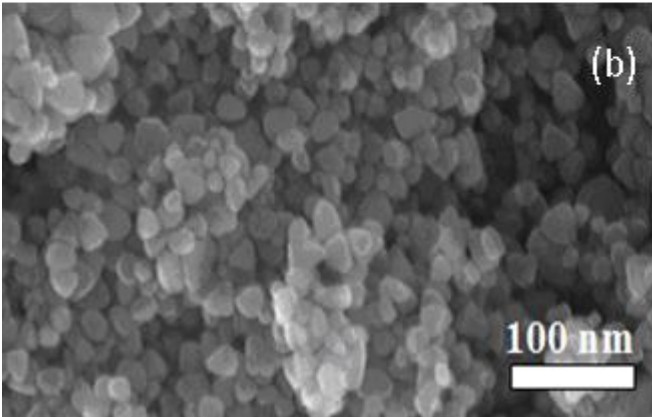

**Figure 4.** TEM photographs: (**a**) ZnO and (**b**) ZnO:V3%.

### 3.3. Antibacterial Activities

The bacterial envelope is a complex, multi-layered structure that maintains the morphology of the cell as well as the osmotic pressure of the cytoplasm and provides rigidity that protects it against mechanical forces. It is composed mainly of peptidoglycans (essentially amino acids and sugars). Gram$^+$ bacteria have a membrane rich in peptidoglycans forming a thick layer (20–80 nm); on the other hand, Gram$^-$ bacteria are composed of two plasma membranes: the outer membrane and the plasma membrane with a thin layer of peptidoglycan forming a thickness of 7–8 nm [39,40].

The antimicrobial activities of pure and vanadium-doped ZnO nanoparticles against two Gram$^+$ bacteria (*S. aureus* and *B. cereus*), two Gram$^-$ bacteria (*E. coli* and *P. aeruginosa*), and two fungi (*C. albicans* and *R. glutinis*) were investigated via the inhibition zone (IZ), as shown in Figure 5.

The antibacterial and antifungal activities of pure ZnO and V-doped NPs are given in Table 3. It is clear from the table that ZnO NPs doped with 1 at.% and 3 at.% V exhibited higher antibacterial activities against *S. aureus*, *E. coli,* and *P. aeruginosa* and higher antifungal activity for *C. albicans* compared to 5 at.% loading and pure ZnO. This result could be affected by the electronic defects created in the ZnO network after the incorporation of vanadium impurities. In addition, the particle size exhibited a crucial role in bactericidal and/or bacteriostatic mechanisms. In fact, as a result of reducing crystallite size, an increase in specific surface area occurs, followed by an enhancement of particle surface reactivity. All these phenomena led to an increase in the inhibition zone. As an example, the variation of *E. coli* bacteria's inhibition zone with particle size is presented

in Figure 6. For 26 nm, the inhibition zone IZ of *Escherichia coli* bacteria was 15 mm, and when the crystallite size is increased, the IZ is reduced to 12 mm. The obtained results are in concordance with the works done by Zhang and his colleagues, Yamamoto's team, and Sawai et al. [41–43].

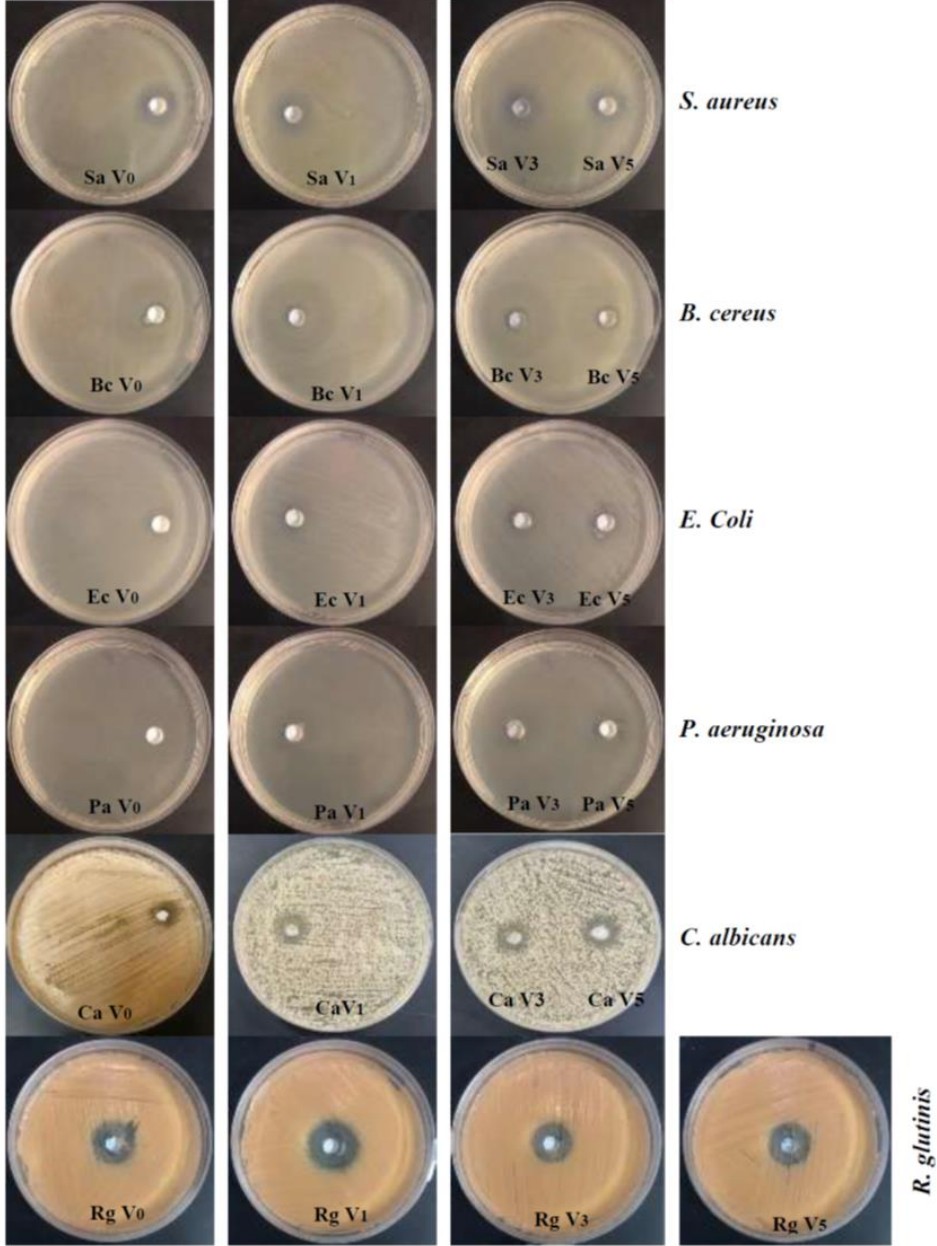

**Figure 5.** Inhibition zones of vanadium doped ZnO against *Staphylococcus aureus*, *Bacillus cereus*, *Escherichia coli*, *Pseudomonas aeruginosa*, *Candida albicans*, and *Rhodotorula glutinis* for control pure ZnO (V0), vanadium doped ZnO, V1-ZnO (1 at.%), V3-ZnO (3 at.%), and V5-ZnO (5 at.%).

In contrast, and as illustrated in Table 3 and Figure 7, it is well noted that for the ZnO:$V_{1\%}$ sample, the inhibition zone IZ of *P. aeruginosa* bacteria has been reduced from 20 nm (for the undoped ZnO sample) to 15 nm (for the 1 at.% doped one). Despite the lowest average crystallite size of the ZnO:$V_{1\%}$ sample (26 nm) in comparison with that of pure ZnO (31 nm), a reduction of the inhibition zone was observed for the above-mentioned bacteria (*P. aeruginosa*). So, we can say that there is another reason for this behavior. It is probably due to the non-solubility of the ZnO:$V_{1\%}$ nanopowders in DMSO solvent.

**Table 3.** Inhibition zones diameters showing antimicrobial activities of ZnO and V modified ZnO against two Gram$^+$ and two Gram$^-$ bacteria and two fungi.

| Samples | Antimicrobial Activity (Inhibition Zone mm) | | | | | |
| --- | --- | --- | --- | --- | --- | --- |
| | Gram$^+$ Bacteria | | Gram$^-$ Bacteria | | Fungi | |
| | *S. aureus* | *B. cereus* | *E. coli* | *P. aeruginosa* | *C. albicans* | *R. glutinis* |
| V0ZO | 14 | 20 | 12 | 20 | 14 | 25 |
| V1ZO | 16 | 20 | 15 | 15 | 15 | 27 |
| V3ZO | 15 | 20 | 15 | 22 | 16 | 20 |
| V5ZO | 15 | 20 | 14 | 22 | 15 | 20 |

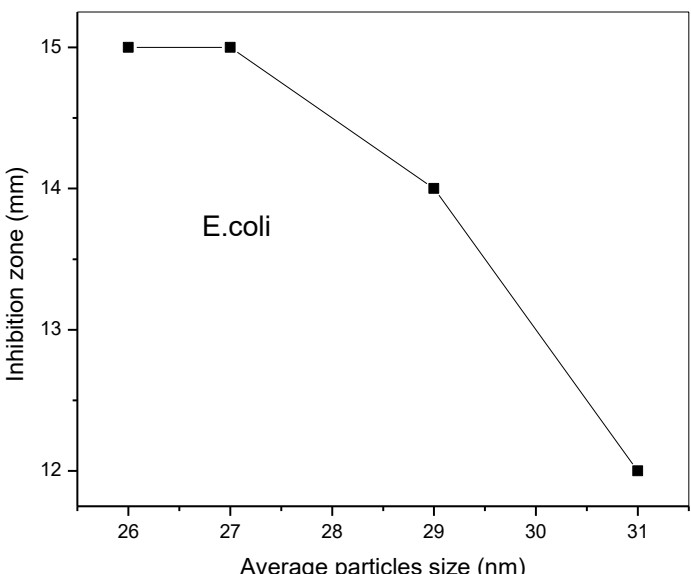

**Figure 6.** Variation of E. coli inhibition zone with particle size.

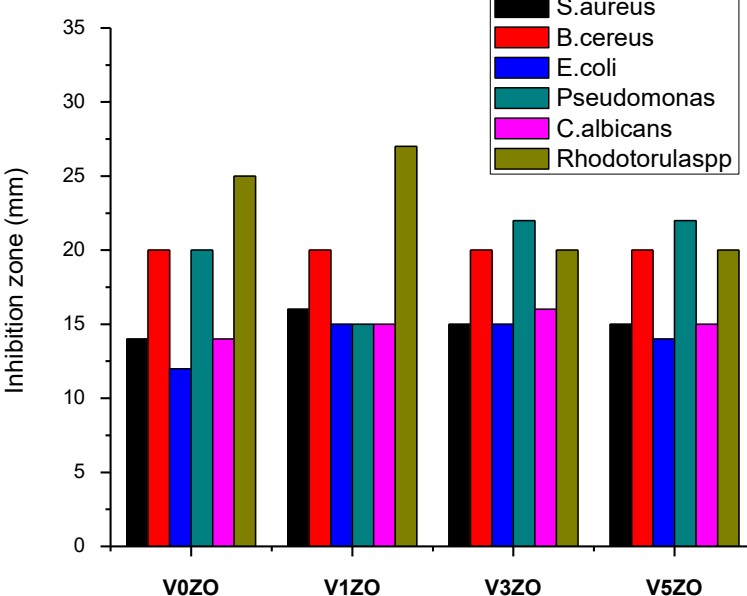

**Figure 7.** Inhibition zone diameter for different synthesized samples, indicating the antibacterial activities against Gram$^+$ and Gram$^-$ bacteria and fungi.

The antibacterial activity is also dependent on nanoparticle shapes [44,45]. The shape dependence is explained in terms of the number of active facets existing in nanoparticles.

The rod shape of zinc oxide had (111) and (100) facets, whereas the sphere shape had mainly (100) facets. In our case, from XRD spectra, V1ZO and V3ZO exhibited a more intensive (100) peak; therefore, they had a larger number of facets, which is the explanation of the high antibacterial activities for both samples compared to the other ones.

Table 4 reports a comparison between the antibacterial activities of our samples and those of other research works cited in the literature. These results are in concordance with the work performed by Danial and his team [16], which showed an enhancement of inhibition zones when indium ions are incorporated in the network of material. Our results were also well corroborated by Kayani et al. [31], who showed a high diameter of the inhibition zone of ZnO: B towards *E. coli* bacteria. Laraib's team has also noticed the high influence of magnesium dopant on ZnO antibacterial activities [46]. As illustrated in Table 3 and Figure 7, most antibacterial activity studies emphasized that metallic dopants caused an enhancement in the inhibition zones of the studied bacteria. Despite the very low nanomaterial concentration (10 µg/mL) of our samples compared to those of other samples, they exhibited high activity against microbes. In fact, in all the scientific works noted in Table 4, the authors have utilized a high concentration of the nanomaterials suspended in dimethyl sulfoxide (DMSO) compared to the concentration of V-doped ZnO that has been used in the present work. This is considered a strong point in our work.

**Table 4.** Comparison of antibacterial activities for some doped ZnO nanomaterials.

| Samples | Concentration | Inhibition Zone (mm) | | Reference |
|---------|---------------|-----------------------|---|-----------|
| | | *E. coli* | *S. aureus* | |
| ZnO:In | 10 mg/mL | 16 | 12 | [26] |
| ZnO:Mg | 90 µg/mL | 19 | 15 | [46] |
| ZnO:Cu | 50 µg/mL | 23 | 22 | [47] |
| ZnO:V | 1 mg/mL | 22 | - | [48] |
| ZnO:Cr | 2 mg/mL | 13 | 15 | [49] |
| ZnO:V | 10 µg/mL | 15 | 16 | This work |

*3.4. Antimicrobial Activities Mechanisms*

The antimicrobial activity mechanisms of ZnO material are $Zn^{2+}$ release, electrostatic interactions, and ROS species production. The reactive oxygen species ROS are highly produced by aquatic ZnO nanoparticle suspensions. Numerous scientific studies have confirmed the key role of ROS in the antibacterial activity mechanism [50,51]. In our study, the ROS species are created without UV exposure but are generated in visible conditions. The sources of free electrons were the interstitial Zn (Zni) defects that are negatively charged, and the sources of free holes were zinc vacancies (VZn) defects that are positively charged. So, in the absence of UV light, the existence of electronic defects in the ZnO network exhibited a key role in electron-hole pair production and therefore the production of ROS species that are very important in the enhancement of ZnO antimicrobial activities. This is in accordance with numerous scientific studies [52]. As mentioned above, V-ZnO samples exhibited higher activity against most bacteria and fungi compared to pure ones. This enhancement can be explained as follows: the addition of vanadium to the ZnO network increases the light absorption of ZnO in the visible range, so a strong photogeneration of electron-hole pairs occurs, which leads to ROS production. These species (superoxide, hydrogen peroxide, and hydroxide), by their toxicity, destroyed bacteria's components such as lipids, proteins, and DNA when they crossed the cell membranes of bacteria.

## 4. Conclusions

A low-cost chemical technique named sol-gel has been used to synthesize pure V-ZnO nanoparticles with three different concentrations (1, 3, and 5 at.%) under supercritical dry conditions (ethyl alcohol as co-solvent; Tc = 243 °C; Pc = 63.6 bars). All the samples are well crystallized in a hexagonal Wurtzite structure. The average crystallite size was estimated using Sherer's formula. Its value was 31 nm for the undoped ZnO, and it decreased

to 26 nm after doping with vanadium at 1 at.%. Pure and doped nanoparticles had spherical and quasi-spherical shapes. Due to the reduction of particle size with vanadium incorporation, V-doped samples exhibited high antimicrobial activities against Gram$^+$ and Gram$^-$ bacteria and fungi. The antimicrobial activity mechanisms of ZnO material are Zn$^{2+}$ release, electrostatic interactions, and ROS species production. The production of ROS species due to light absorption in the visible range and the thermal excitation of electronic defects is a key parameter for bactericidal and/or bacteriostatic mechanisms. Finally, we could consider ZnO:V materials a promising antimicrobial agent in the visible region of light due to their influence on minimizing the band gap of ZnO.

**Author Contributions:** Writing and editing, L.A.; investigation, A.M.S.; conceptualization, A.A.A.; formal analysis, M.H.K.A.; formal analysis, N.H.A.; data curation, writing and editing, supervision, M.H. All authors have read and agreed to the published version of the manuscript.

**Funding:** This research received no external funding.

**Institutional Review Board Statement:** Not applicable.

**Data Availability Statement:** All the data from this study are included in the manuscript. The data used to support the findings of this study are available from the corresponding author upon request.

**Conflicts of Interest:** Authors declare that there is no conflict of interest.

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
