# Peer review of "ZnO:V Nanoparticles with Enhanced Antimicrobial Activities"

_jcs, doi:10.3390/jcs7050190_

Round 1

Reviewer 1 Report

Reviewers' comments:

The authors prepared the V-ZnO nanoparticles with different vanadium concentrations through the sol-gel approach. The as-synthesized V-doped ZnO exhibited the positive influence on bacteria and fungi growth inhibition. However, some problems remain in the paper. I recommend that major revision of the manuscript will be required.

(1)    Compared with reported doped-ZnO nanoparticles, the advantage of V-ZnO nanopowders should be clarified, which is important to better understand what the paper described. In my opinion, the novelty of study must be clearly addressed in the section of “Introduction” before this article can be considered for publication in journal.

(2)    I want the author should provide the clear evidence to confirm the introduction of Zr into ZnO nanoparticles. Meanwhile, the concentration of doped Zr species should be measured by ICP.

(3)    I suggest that the authors should explain why the high antimicrobial activity was observed after V doping. More details in the contents of produced ROS species and how ROS enhance the antimicrobial activity should be provided.

(4)    There are some grammatical issues that, if corrected, would greatly improve the readability of the paper.

Author Response

Dear Editors and Reviewers,

On behalf of my colleagues, I want to thank the editors and the reviewers for all the comments and recommendations. Those comments are all valuable and very helpful for revising and improving our paper. We have modified the main text accordingly, and detailed corrections are listed in the following manuscript, point by point. All the issues raised by the Reviewers were answered and highlighted in the revised manuscript to allow easy reading for the reviewer/editor. English Language of the whole paper has been improved, and the revised manuscript has been proofread before submission.

Dr. Mokhtar Hjiri

Response to Reviewer 1 Comments:

The authors prepared the V-ZnO nanoparticles with different vanadium concentrations through the sol-gel approach. The as-synthesized V-doped ZnO exhibited a positive influence on bacteria and fungi growth inhibition. However, some problems remain in the paper. I recommend that major revision of the manuscript be required. 

Q 1: Compared with reported doped-ZnO nanoparticles, the advantage of V-ZnO nanopowders should be clarified, which is important to better understand what the paper described. In my opinion, the novelty of study must be clearly addressed in the section of “Introduction” before this article can be considered for publication in journal.

Response 1: The authors would like to thank the reviewer for taking the time to read the manuscript. The advantages of the usage of vanadium as dopant for the antibacterial activities of ZnO is clarified; a paragraph in this context is added in the text. The novelty of this study is also added.

Q 2: I want the author to provide the clear evidence to confirm the introduction of Zr into ZnO nanoparticles. Meanwhile, the concentration of doped Zr species should be measured by ICP.

Response 2: Thank you dear reviewer for this good remark. By using Energy Dispersive X-ray technique (EDX) we have confirmed the incorporation of V-dopant in the ZnO network (Please see Figure 3.b). This technique leads to highlight the vanadium concentration incorporated in ZnO network. Please see in the text, we have added a table shown the exact percentages of different elements of V3ZO materials.

Q 3: I suggest that the authors should explain why the high antimicrobial activity was observed after V doping. More details in the contents of produced ROS species and how ROS enhance the antimicrobial activity should be provided.

Response 3: We appreciate the reviewer's attention. In fact, the production of reactive oxygen species is an main parameter for the antibacterial activities machanism. V-ZnO samples exhibited higher activities against bacteria and fungi compared to ture one. This enhancement can be explained as follow: the addition of vanadium to ZnO network increases the light absoption of ZnO in the visible range; so a strong photogeneration of electrons-holes pairs occurs; and this led to ROS production without using UV irradiations. These reactive oxygen species (ROS) (superoxide, hydrogen peroxide and hydroxide), leads to the destuction of bacteria’s components like lipids, proteins, and DNA, after its crossing cells membranes of bacteria.

    This part is added in the manuscript.

Q 4: There are some grammatical issues that, if corrected, would greatly improve the readability of the paper.

Response 4: The manuscript is revised and grammatical mistakes and issues were corrected. The authors appreciate the reviewer's thoughtful comments that have significantly improved our manuscript.

Reviewer 2 Report

1- instead of G+ and G-  use gram+ and gram-

2- . Furthermore, its great optical and electrical properties lead 54 this material to find several applications in different fields like photocatalysis, solar cells, 55 gas sensors, optoelectronic devices, …etc [13-16].  What does this sentence have to do with antibacterial properties, out of the contest?

3- what is the application of such nanoparticles - the bigger picture, missing in the introduction.

4 additionally state of the art is missing, can you mention what our studies have done so far

Author Response

Dear Editors and Reviewers,

On behalf of my colleagues, I want to thank the editors and the reviewers for all the comments and recommendations. Those comments are all valuable and very helpful for revising and improving our paper. We have modified the main text accordingly, and detailed corrections are listed in the following manuscript, point by point. All the issues raised by the Reviewers were answered and highlighted in the revised manuscript to allow easy reading for the reviewer/editor. English Language of the whole paper has been improved, and the revised manuscript has been proofread before submission.

Dr Mokhtar Hjiri

Responses to Reviewer 2 Comments

Q 1: instead of G+ and G-  use gram+ and gram-

Response 1: We apologize for this confusion, we have corrected those sentences. Thank you for the careful reading. In all the text, G+ and G- are replaced by gram+ and gram-

Q 2: Furthermore, its great optical and electrical properties lead 54 this material to find several applications in different fields like photocatalysis, solar cells, 55 gas sensors, optoelectronic devices, …etc [13-16].  What does this sentence have to do with antibacterial properties, out of the contest?

Response 2: I agree with the respected reviewer. We have to mention the antibacterial activities application. We have cancelled this part and it was compensated by the antibacterial application.

Q 3: what is the application of such nanoparticles - the bigger picture, missing in the introduction.

Response 3: We thank the reviewer for this significant remark. More details about our goal which is to study the effect of vanadium addition on the antimicrobial activities application has been added to the introduction part in order to avoid any confusion.

Q 4:  additionally state of the art is missing, can you mention what our studies have done so far

Response 4: More sentences on sate of art has been added in the introduction section in order to be more correct. We appreciate the Editors' and Reviewers’ gracious work earnestly, allowing us to enhance and improve our manuscript, and hope that the availed improvements will meet their approval.

Once again, thank you very much for your pertinent comments, recommendations, and advice.

Reviewer 3 Report

Dear Editor and Authors,

Nanomaterial based approaches are for sure among the most promising approaches to defeat development of resistant bacterial infections and fungal infections. Testing similar materials by many times will definitely in different labs is critical to assess their reliability and exact potential. In this manner, this study is valuable by comparing the response of gram - and gram + along with fungi to ZnO nanoparticles. Before the paper reaches the readers, there should be some improvements.

1- The language must be checked-for instance there are "tense choice" mistakes, grammatical mistakes, microorganism names (must be italic) etc.

2- EDx images should be improved.

3- There is no experimentation to support "Antimicrobial activities mechanisms" section.

4- Antimicrobial activity discussions must be expanded through the discussions on how surface chemistry of the microbial species possibly played role during the nanoparticles' tocixities.

Kind regards,

Round 2

Reviewer 1 Report

The author has provided good answers to all the questions, and the manuscript has been revised according to the comments. 

Reviewer 2 Report

I would like to thank the auothers for taking care of all the reviewer's concerns

Reviewer 3 Report

Dear Editor,

In this form, the antibacterial activity is now supported by the literature at acceptable level. The other improvements also have made the paper ready.

Kind regards,